# Role of MalQ Enzyme in a Reconstructed Maltose/Maltodextrin Pathway in *Actinoplanes* sp. SE50/110

**DOI:** 10.3390/microorganisms12061221

**Published:** 2024-06-18

**Authors:** Camilla März, Sophia Nölting, Lars Wollenschläger, Alfred Pühler, Jörn Kalinowski

**Affiliations:** 1Microbial Genomics and Biotechnology, Center for Biotechnology, Bielefeld University, 33615 Bielefeld, Germany; cmaerz@cebitec.uni-bielefeld.de (C.M.); noelting@cebitec.uni-bielefeld.de (S.N.); lwollens@cebitec.uni-bielefeld.de (L.W.); 2Senior Research Group in Genome Research of Industrial Microorganisms, Center for Biotechnology, Bielefeld University, 33615 Bielefeld, Germany; puehler@cebitec.uni-bielefeld.de

**Keywords:** *Actinoplanes*, acarbose, acarviosyl metabolites, α-1,4-glucan, maltose/maltodextrin pathway, carbohydrate, MalQ

## Abstract

The pseudotetrasaccharide acarbose, produced by *Actinoplanes* sp. SE50/110, is a relevant secondary metabolite used in diabetes type II medication. Although maltose plays a crucial role in acarbose biosynthesis, the understanding of the maltose/maltodextrin metabolism and its involvement in acarbose production is at an early stage. Here, we reconstructed the predicted maltose–maltodextrin pathway that involves four enzymes AmlE, MalZ, MalP, and MalQ. An investigation of enzyme activities was conducted through in vitro assays, leading to an expansion of previously postulated substrate spectra. The maltose-induced α-glucosidase AmlE is noteworthy for its high hydrolysis rate of linear α-1,4-glucans, and its capability to hydrolyze various glycosidic bonds. The predicted maltodextrin glucosidase MalZ showed slow hydrolysis activity on linear α-glucans, but it was resistant to acarbose and capable of releasing glucose from acarbose. AmlE compensates for the low activity of MalZ to ensure glucose supply. We determined the enzyme activity of MalP and its dual function as maltodextrin and glycogen phosphorylase. The 4-α-glucanotransferase MalQ plays a central role in the maltose/maltodextrin metabolism, alongside MalP. This study confirmed the simultaneous degradation and synthesis of long-chain α-glucans. The product distribution showed that with an increasing number of glycosidic bonds, less glucose is formed. We found that MalQ, like its sequence homolog AcbQ from the acarbose biosynthetic gene cluster, is involved in the formation of elongated acarviosyl metabolites. However, MalQ does not participate in the elongation of acarbose 7-phosphate, which is likely the more readily available acceptor molecule in vivo. Accordingly, MalQ is not involved in the formation of acarviosyl impurities in *Actinoplanes* sp. SE50/110.

## 1. Introduction

*Actinoplanes* sp. SE50/110 (ATCC 31044), an aerobic Gram-positive spore-forming actinobacterium, produces the α-glucosidase inhibitor acarbose (acarviosyl-maltose). Acarbose is a pseudo-tetrasaccharide that blocks the active site of α-glucosidases, which inhibits the cleavage of polysaccharides. This reduces the rise in blood glucose levels after the consumption of starch-containing meals. Due to its ability to reduce postprandial hyperglycemia, acarbose has been utilized as an active ingredient in the treatment of diabetes mellitus type II [1,2]. In its natural habitat, this characteristic putatively provides a competitive advantage in food competition because the rival enzymes necessary for carbohydrate uptake are inhibited by acarbose, while its own secreted α-glucosidases are resistant [3]. Thus, the hypothesis is that acarbose 7-phosphate is more abundant in cells than acarbose [4].

The present understanding of carbohydrate metabolism in *Actinoplanes* sp. SE50/110 relies mainly on a model founded through homology comparisons with other model organisms and current gene annotations [5,6,7]. The disaccharide maltose plays a unique role in the culture media as it serves as the primary carbon source, providing energy and serving as a key precursor in acarbose biosynthesis [8]. The interplay of carbohydrates plays a crucial role in the build-up and degradation of higher maltodextrins, which is essential for the formation of glycogen in *Actinoplanes* sp. SE50/110. However, this understanding is in its infancy.

The metabolism of α-1,4-linked glucose polymers in the maltose/maltodextrin system has been extensively researched in multiple microorganisms [9,10,11]. The *mal* genes in *Escherichia coli* (*E. coli*) are regulated by MalT*^Ec^* (MalT in *E. coli*), a transcriptional activator that is induced by maltotriose and ATP and is crucial for the expression of the *mal* genes [11]. However, in *Actinoplanes* sp. SE50/110 the maltose-dependent regulator MalT*^As^* (MalT sequence homolog in *Actinoplanes* sp. SE50/110) is not responsible for regulating genes involved in maltose metabolism [12].

In *E. coli* the most important roles are determined by the interplay of MalQ*^E^*^c^ (MalQ in *E. coli*) and MalP*^Ec^* (MalP in *E. coli*) [11]. MalQ is a 4-α-glucanotransferase, that cleaves linear maltodextrins. The non-reducing maltodextrinyl moiety of the donor molecule is transferred onto another maltodextrin while the remaining reducing sugar end is released. According to recent studies, MalQ*^Ec^* also converts short maltodextrins, such as maltose, to longer maltodextrins, whereas previous studies presumed that maltotriose was the smallest possible substrate [9,13]. In other organisms, such as *E. coli* or *Corynebacterium glutamicum*, the maltodextrin phosphorylase MalP*^Ec/Cg^* (MalP in *E. coli*, MalP in *C. glutamicum*) plays a significant role alongside MalQ*^Ec/Cg^* [10,14]. The activity of both enzymes complements each other, enabling efficient utilization of maltodextrins of different lengths. MalP*^Ec/Cg^* phosphohydrolyzes the α-glucans built up by MalQ*^Ec/Cg^*, resulting in glucose 1-phosphate that is available for glycolysis. In many bacteria [11], a second α-glucan phosphorylase, encoded by the *glgP* gene, has been characterized as glycogen phosphorylase. In *E. coli*, the catalytic domains of both isoenzymes remain conserved, although variations have been detected in substrate preferences and regulation [15]. Although both enzymes share the same catalytic mechanism, variations in substrate specificity were identified in *E. coli* [16]. GlgP*^Ec^* (GlgP in *E. coli*) catalyzes glycogen breakdown from outer chains, stops three to four residues from the first α-1,6 branching point and thereby generates a phosphatase-limited dextrin (pl dextrin) which is linearized by the debranching enzyme GlgX*^Ec^* (GlgX in *E. coli*) [14,17]. Linear maltodextrins can serve as substrates for other building or degrading enzymes.

Different regulatory mechanisms were discovered for *malP^Cg^* and *glgP^Cg^* genes in *C. glutamicum*. GlgP*^Cg^* (GlgP in *C. glutamicum*) is constitutively expressed, while *malP^Cg^* expression is dependent on the carbon source [18]. *Streptomyces glaucescens* GLA.O and *Streptomyces lividans* TK23, which are taxonomically more closely related to *Actinoplanes* sp., also lack the second α-glucan phosphorylase gene [6]. This suggests a different role in the breakdown of maltodextrins and glycogen within this taxon [6].

The maltose-induced α-glucosidase AmlE was deemed crucial for maltose metabolism in *Actinoplanes* sp. SE50/110 based on in vivo studies and functional characterization assays [6]. Furthermore, it is controlled by the AmlR transcriptional regulator. The hydrolase activity of AmlE was detected in raw protein extract assays in the wildtype for different α-1,4-glucans, resulting in the release of glucose [6]. However, the absence of AmlE in a gene deletion mutant cannot be compensated by the activity of putative MalZ*^As^* (MalZ in *Actinoplanes* sp. SE50/110) and MalQ*^As^* (MalQ in *Actinoplanes* sp. SE50/110) proteins, leading to an inadequate supply of glucose [6]. Interestingly, an *amlE* homolog was identified within the *gac* acarbose biosynthetic gene cluster of S. *glaucescens* GLA.O. The precise categorization of AmlE within maltose/maltodextrin metabolism and the connection between acarbose and carbohydrate metabolism therefore remain unresolved.

Although acarbose is the main product, *Actinoplanes* sp. SE50/110 also produces several acarviosyl metabolites, here referred to as impurities [2,19,20]. All homologs have the acarviosyl unit consisting of C7-cyclitol and 4-amino-4,6-dideoxyglucose as the elementary core structure and differ in the attached sugar moieties [21,22]. The inhibitory spectrum varies with length. While low molecular weight pseudo-oligosaccharides tend to inhibit maltases and disaccaridases, longer homologs are more effective against α-amylases [2,21]. In 2014 [8], the dependence of the provided carbon source and the formation of further acarviosyl metabolites were demonstrated. The variable saccharide moiety is supplied by the carbon source in the medium, imported and linked to the acarviosyl core structure [8]. For the minor metabolites, the saccharide component must be produced intracellularly during maltose/maltodextrin and glycogen metabolism [8].

The objective of this study is to functionally characterize enzymes that may be involved in maltose/maltodextrin metabolism through sequence homology comparisons, heterologous expression in *E. coli* and enzyme assays.

## 2. Materials and Methods

### 2.1. Bacterial Strains, Construction of Expression Vectors, and Culture Conditions

The bacterial strains utilized were *E. coli* DH5α for plasmid construction and, for protein production, *E. coli* BL21 (DE3) pLysS (Table 1). The pJOE5751.1 vector, containing an N-terminal His_6_-tag, was isolated with a GeneJET Plasmid Miniprep Kit (Thermo Fisher Scientific, Waltham, MA, USA) for inducible gene expression. The coding sequence (CDS) of each target gene to produce heterologous proteins was amplified via polymerase chain reaction (PCR) using genomic DNA isolated from *Actinoplanes* sp. SE50/110. PCR products were purified from an agarose gel using the GeneJET Gel Extraction Kit (Thermo Fisher Scientific) and then cloned into the expression vector pJOE5751.1 using the Gibson Assembly (protocol from New England Biolabs, Ipswich, MA, USA). The assembled plasmids were transformed into *E. coli* DH5α via heat shock and plated on selective LB agar plates (100 µg mL^−1^ ampicillin (Amp)) and incubated at 37 °C. The plates were screened for positive transformants, and the nucleotide sequence was confirmed by sequencing. For high-efficiency protein production, the verified plasmids were transformed into *E. coli* BL21 (DE3) pLysS.

### 2.2. Heterologous Protein Expression

Pre-cultures of *E. coli* BL21(DE3) pLysS were grown in 10 mL of LB (100 µg mL^−1^ Amp, 25 µg mL^−1^ chloramphenicol (Cm)) overnight at 37 °C and 200 rpm. For gene expression, 50–200 mL of LB (100 µg mL^−1^ Amp, 25 µg mL^−1^ Cm) was inoculated to an OD_600_ of 0.1. When an OD_600_ of 0.6–0.8 was reached, the temperature was set to 16 °C. To induce the protein expression L-Rhamnose (f.c., 0.2%) was added and the cultivation was continued for 16–20 h. Subsequently, the cells were harvested by centrifugation (10 min, 5500× *g*, 4 °C) and the supernatant was discarded. The cell pellets were resuspended in ice-cold 1× LEW buffer (Protino Ni-TED kit, Macherey-Nagel, Düren, Germany).

The cells were disrupted in tubes filled with Zirconia beads (0.1 and 0.5 mm in size, Carl Roth, Karlsruhe, Germany) using a homogenizer (Precellys 24 homogenizer, Bertin Technologies, Montigny le Bretonneux, France) with three cycles of 6500 rpm for 30 s and cooling intervals of 5 min on ice in between. A centrifugation step was performed to remove cell debris and beads (20 min, 21,000× *g*, 4 °C). The supernatants were transferred to a fresh 2 mL reaction tube and centrifuged to remove the remaining Zirconia beads. The crude extracts were applied to Protino^®^ Columns and purification steps were performed according to the manual (Protino Ni-TED kit, Macherey-Nagel, Düren, Germany). Elution was performed in four steps of 500 µL each. The fractions were collected, and protein identities were confirmed by SDS-PAGE and MALDI-TOF-MS/MS analysis. For protein quantification according to the Bradford method, 1x Roti^®^-Quant reagent was utilized (Carl Roth, Sigma Aldrich (St. Louis, MI, USA)).

### 2.3. Substrates and Standards

For in vitro enzyme assays and analytical measurements different linear α-glucans were used as substrates and standards. Glucose (G1), maltose (G2), maltotriose (G3), sucrose (Suc), trehalose (Tre), and α-D-Glucose 1-phosphate were acquired from Carl Roth, Sigma Aldrich (St. Louis, MI, USA), or VWR (Radnor, PA, USA). Maltotetraose (G4), maltopentaose (G5), maltohexaose (G6), maltoheptaose (G7), maltooctaose (G8), and isomaltose (Iso) were purchased from Megazyme (Bray, Wicklow, Ireland). Acarbose (Acb) was manufactured and kindly provided by Bayer AG (Leverkusen, Germany). Glycogen from oysters was purchased from Thermo Fisher Scientific (Waltham, MA, USA). Acarbose 7-phosphate (Acb-7P) was synthesized from acarbose and ATP using AcbK as described by Nölting et al. [4].

### 2.4. In Vitro Enzyme Assays

A modified enzyme assay based on the protocol of Seibold et al. (2009) [10] was used to monitor all enzyme activities.

The 4-α-glucanotransferase activity of MalQ*^As^* was determined by the production of shortened and elongated α-glucans from a range of substrates (G2–G8, Acb, Acb-7P, Acb + G2, Acb-7P + G2). The products were analyzed by a continuous hexokinase/glucose 6-phosphatedehydrogenase assay, thin-layer chromatography (TLC), high-performance anion exchange chromatography (HPAEC) with pulsed amperometric detection (PAD) analysis (HPAEC-PAD), and liquid chromatography–electrospray ionization mass spectrometry (LC-ESI-MS) analysis as described below.

To determine the initial substrate preferences of MalQ*^As^*, the rate of glucose release from different substrates was measured spectrophotometrically [10]. Each reaction mixture consisted of 50 mM potassium phosphate buffer (pH 7.0), 25 mM MgCl_2_, 2 mM ATP, 2 mM NADP, 2 U hexokinase (Sigma Aldrich), 2 U glucose-6-phosphate dehydrogenase (Alfa Aesar by Thermo Fisher Scientific), and 5 mM of the corresponding substrate. The reaction mixtures without MalQ*^As^* enzyme were pre-incubated in a 96-well plate (flat-bottom Nunc^TM^ 96-Well Polystyrene Plates from Thermo Scientific, Waltham, MA, USA) at 30 °C and the absorption at 340 nm was measured in a Tecan Infinite M200 microplate reader using i-control 10.1 software (Tecan Group AG, Männedorf, Switzerland). When there were no more fluctuations in the signal, MilliQ and purified enzyme were added to 200 µL to start the reaction. The released glucose was phosphorylated to glucose-6P by hexokinase. Glucose-6P served as a substrate for glucose-6P dehydrogenase and was oxidized to 6-phosphogluconolactone, while NADP was reduced to NADPH. The absorption maximum at 340 nm was measured. The increasing absorption corresponds to the equimolar amount of glucose released by the reduction of NADP^+^ to NADPH [10].

Glucose was used as the positive control and calibration curve. All samples were assayed in triplicate. The initial rate was calculated from the linear increase in the absorption during the first ten minutes.

The maltodextrin glucosidase activity of MalZ*^As^* and the α-glucosidase activity of AmlE were monitored by glucose release from different α-glucans (G2, G3, G4, G5, G8) at 30 °C by the hexokinase/glucose 6-phoshpate dehydrogenase assay as described above. The enzyme concentrations used were adjusted to 1 µM AmlE and 4 µM MalZ*^As^* for each substrate preference assays. Acarbose and acarbose 7-phosphate were used independently or in combination with maltose or maltotriose as substrates with 2 µM enzyme each, to assess the potential inhibitory effects of acarbose on the endogenous α-glucosidases.

Analogously, the activity of the α-glucan phosphorylase MalP*^As^* was measured by an phosphoglucomutase/glucose 6-phoshpate dehydrogenase assay [18]. Due to the phosphorolytic activity of 1 µM of purified MalP*^As^*, glucose 1-phosphate was released from different α-glucans. The activity of a phosphoglucomutase converts glucose 1-phosphate to glucose 6-phosphate, which serves as a substrate for glucose 6-phosphate dehydrogenase, while NADP^+^ is reduced to NADPH. As substrates, α-glucans of different lengths, ranging from maltose to maltooctaose, were tested.

### 2.5. Visualization of Enzyme Assay Products by Thin-Layer Chromatography (TLC)

TLC was used to differentiate the products formed in enzyme assays. All reaction mixtures contained 50 mM of potassium phosphate buffer (pH 7.0), 25 mM of MgCl_2_, and 10 mM of α-glucan (G2, G3, G4, G5, G6, G7, G8) as a substrate in a total reaction volume of 20 µL. The reaction was initiated by adding 10 µM of purified enzyme and incubated at 30 °C for 5 h. For MalP assays, 5 mM of ATP was added. Silica gel 60 F254 plates (Sigma Aldrich) were used as the stationary phase and a mixture of 1-butanol, 2-propanol, ethanol, and water (3:2:3:2) as the mobile phase [25]. To visualize the separated product spots, the silica plate was sprinkled with 4% (*w*/*v*) sulfuric acid in methanol and heated at 150 °C for 1 min until the darkened spots appeared.

### 2.6. HPAEC-PAD Analysis of MalQ^As^ Reaction Products

For high-performance anion exchange chromatography (HPAEC) with pulsed amperometric detection (PAD) analysis, MalQ*^As^* samples were processed as described above for the TLC analysis. The Dionex ICS-6000 HPIC system (Thermo Fisher Scientific) was used for the quantification of α-1,4-glucans according to Nölting et al. (2023) [4]. The samples were diluted 40-fold and 400-fold and a total volume of 20 µL was injected. At a flow rate of 1 mL min^−1^, eluent A (166 mM of ammonium hydroxide) and eluent B (1 M of sodium acetate with 166 mM of ammonium hydroxide) were used with the following gradient: 6.5 min 10% B, 31.5 min 25% B, 34.0 min 25% B, and 44.0 min 10% B. For separation of linear α-glucans, a Dionex CarboPac PA100 column (250 × 4 mm, 8.5 µm, Thermo Fisher Scientific) coupled to a Dionex CarboPac PA100 Guard column (250 × 4 mm, 8.5 µm, Thermo Fisher Scientific) were used. Pulsed amperometric detection was performed at a system temperature of 30 °C using the gold, carbo, quad waveform with a non-disposable gold electrode and a AgCl reference electrode. Data evaluation was based on Chromeleon Chromatography Data System 7.2.10 software (Thermo Fisher Scientific).

### 2.7. Product Identification of MalQ^As^ and MalP^As^ by LC-ESI-MS

For liquid chromatography–electrospray ionization mass spectrometry (LC-ESI-MS) measurement of acarviosyl metabolites, the reaction mixture of MalQ*^As^* contained 10 mM of purified enzyme, 50 mM of potassium phosphate buffer (pH 7.0), 25 mM of MgCl_2_, and 10 mM of substrate mixtures (G3 + Acb, G3 + Acb-7P) in a total volume of 100 µL was incubated at 30 °C for 5 h. Mixtures with heat-inactivated MalQ*^As^* were used as a negative control. To identify glucose 1-phosphate as a release product of MalP*^As^*, 10 mM of ATP and maltoheptaose was added as a substrate. Reaction mixtures with heat-inactivated MalP*^As^* were used as a negative control.

LC-ESI-MS measurements and analyses were performed as described by Nölting et al. (2023) [4]. Mass spectrometry was used for identification of specific enzyme products. The experiment was set up on a micrOTOF-Q hybrid quadrupole/time-of-flight (QTOF) mass spectrometer (MS) equipped with an electrospray ionization (ESI) source (Bruker Daltonics in Billerica, MA, USA). MS was attached to an UltiMate 3000 HPLC system (Thermo Fisher Scientific). Eluent A consisted of an aqueous solution of ammonium formate (10 mM, pH 4.6), whereas eluent B was acetonitrile. The gradient for deposition was as follows: 0 min 80% B, 20 min 15% B, 22.5 min 15% B, 25 min 80% B, and 40 min 80% B. The detection range for MS was set from *m/z* 400–1800 for acarviosyl metabolites in positive ionization mode and for MalP*^As^* products in negative ionization mode. An iHILIC-(P) Classic column (150 × 2.1 mm, 5 μm, Hilicon AB, Umeå, Sweden) was used. Eluent A (20 mM of ammonium bicarbonate, pH 9.3) and eluent B (acetonitrile) were running as follows: 0 min 90% B, 30 min 25% B, 37.5 min 25% B, 45 min 90% B, and 60 min 90% B. A sample volume of 2 µL was injected for each sample and separated at a flow rate of 0.2 µL min^−1^ by an Accucore 150-Amide-HILIC column (150 × 2.1 mm, 2.6 µm, Thermo Fisher Scientific).

Data evaluation was performed with the Compass DataAnalysis 4.2 software (Bruker Daltonics).

## 3. Results

### 3.1. The α-Amylase AmlE Has the Ability to Hydrolyze a Variety of Glycosidic Bonds and Is Sensitive to Inhibition by Acarbose

AmlE (ACSP50_2474) appears to a play central role in the maltose/maltodextrin metabolism in *Actinoplanes* sp. SE50/110. The analysis of protein sequence homology in AmlE reveals several glycoside hydrolase family 13 proteins (EC 3.2.1) in various *Actinoplanes* and *Streptomyces* species. These proteins exhibit high sequence similarity to AmlE, ranging from 67 to 90% identity [6]. Its primary function was identified as maltase, as the absence of AmlE showed that no maltose was metabolized from the medium during growth [6]. In this process, the activity of putative MalZ*^As^* and MalQ*^As^* is not sufficient to compensate for the deletion of AmlE to meet the glucose demand [6].

For a more detailed enzyme characterization, *amlE* was heterologously expressed in *E. coli* BL21(DE3) pLysS and the protein purified via His_6_-tag. Its identity was confirmed by SDS-PAGE and MALDI-TOF analysis. Several α-1,4-linked-glucans with different chain lengths were tested as a substrate (G2, G3, G4, G5 and G8), respectively (Figure 1). The enzymatic activity was analyzed using the hexokinase/glucose-6-phosphate dehydrogenase-coupled enzyme assay. For the determination of substrate preferences, the initial velocity was calculated from the linear increase in absorption during the first minutes of the reaction. Maltotriose was the preferred substrate of AmlE (160 ± 4 µM min^−1^). The initial velocity decreases with further increasing chain length of the substrate (Figure 1).

In different tests using hexokinase/glucose 6-phosphate dehydrogenase coupled enzyme assays, glycogen was also tested as a substrate and a reaction was recorded, leading to the hypothesis that AmlE hydrolyzes free α-1,4-linked glucose but may stop at α-1,6-glycosidic bonds. To evaluate whether α-1,4-linked glucose or α-1,6-glycosidic bonds were cleaved, isomaltose (Isomal, α-1,6-glycosidic bond) was used as a substrate (Figure 1). To determine all possibly hydrolyzed bonds, trehalose (Tre, α-1,1-glycosidic bond) and sucrose (Suc, α-1,2-glycosidic bond) were used as well (Figure 1). Only low amounts of glucose were released using isomaltose as a substrate (0.16 ± 0.03 µM min^−1^) while almost no reaction was recorded for trehalose as a substrate. In contrast, the α-1,2-linkage of sucrose could be cleaved by AmlE (8.38 ± 0.4 µM min^−1^). However, the activity was significantly lower compared to a substrate with an α-1,4-linkage. 

The main product of biotechnological interest of *Actinoplanes* sp. SE50/110 is acarbose. The leading hypothesis is that acarbose provides an advantage in food competition because enzymes necessary for carbohydrate utilization like α-glucosidases in competing organisms are inhibited by acarbose [1,26]. To determine the effect of acarbose on α-glucosidases in *Actinoplanes* sp. SE50/110, AmlE was incubated with the smallest determined substrate, respectively, maltose, acarbose, acarbose 7-phosphate, and combinations thereof (Figure 2). AmlE is characterized by its particularly rapid hydrolysis of α-1,4-glucans. The pseudotetrasaccharide acarbose, on the other hand, is hardly recognized as a substrate as its initial rate is very low (Acb = 0.09 ± 0.02 µM min^−1^). Furthermore, a distinct inhibitory effect of acarbose on AmlE was demonstrated, resulting in a reduced hydrolysis rate by 99.9% when acarbose was combined with maltose (Acb + G2 = 0.12 ± 0.03 µM min^−1^) compared to maltose as a single substrate (G2 = 85.45 ± 2.48 µM min^−1^). Acarbose 7-phosphate as a single substrate was not hydrolyzed. The hydrolysis rate of maltose in combination with acarbose 7-phosphate was decreased by 13% compared to the original rate of maltose. Therefore, acarbose 7-phosphate had a minor effect on the hydrolysis of maltose (Acb-7P + G2 = 74.54 ± 6.47 µM min^−1^).

### 3.2. The Putative Maltodextrin Glucosidase MalZ^As^ Catalyzes the Hydrolysis of Linear Glucans Containing Three or More Glycosyl Units and Uses Acarbose as a Substrate for Slow Release of Glucose

Initial protein sequence homology analysis of the proteins of maltose/maltodextrin metabolism identified ACSP50_4430 as putative maltodextrin glucosidase MalZ*^As^* (EC 3.2.1.20) [6]. This enzyme cleaves single glucose molecules from the reducing end of longer α-glucans with at least three glycosyl units [9,27].

For further specification, the target gene *malZ^As^* was cloned into the inducible vector pJOE5751.1 and heterologously expressed in *E. coli* BL21(DE3) pLysS and purified via His_6_-tag. Protein identity was confirmed by SDS-PAGE and MALDI-TOF-MS/MS analysis.

Since initial studies of maltose/maltodextrin metabolism in *Actinoplanes* sp. SE50/110, both AmlE and MalZ*^As^* have been implicated in the formation of intracellular maltotriose through glycogen degradation [7]. To determine differences in the enzyme activities of AmlE and MalZ*^As^*, the same substrate spectrum was tested for MalZ*^As^*. The enzymatic activity was analyzed using the hexokinase/glucose-6-phosphate dehydrogenase coupled enzyme assay [10]. For determination of substrate preferences G2, G3, G4, G5, and G8 were used. The initial velocity was calculated by determining the linear increase in absorption during the first few minutes of the reaction (Appendix A).

Apart from the generally low hydrolysis rate, the MalZ*^As^* enzyme showed a higher hydrolysis rate for acarbose than for maltotriose (Acb = 0.29 ± 0.02 µM min^−1^, G3 = 0.23 ± 0.03 µM min^−1^) (Figure 2). The hydrolysis rate of the substrate combination of maltotriose and acarbose was increased by 35% compared to maltotriose as the only substrate (G3 + Acb = 0.31 ± 0.02 µM min^−1^). Acarbose 7-phosphate did not lead to a release of glucose and therefore was not used as a substrate. Nevertheless, acarbose 7-phosphate slowed down the hydrolysis of maltotriose by 65% (G3 + Acb 7-P = 0.08 ± 0.03 µM min^−1^).

MalZ*^As^* has been shown to be resistant to inhibition by acarbose and can also use acarbose as a substrate, releasing glucose (Figure 2).

### 3.3. Actinoplanes sp. MalP^As^ Enzyme Has a Dual Function as Maltodextrin and Glycogen Phosphorylase in Glycogen Metabolism

Previous studies of crude protein extracts from *Actinoplanes* sp. SE50/110 have not detected specific MalP*^As^* (maltodextrin phosphorylase) enzyme activity [6]. Interestingly, a sequence homology for a putative maltodextrin phosphorylase (ACSP50_6911; EC 2.4.1.1) was found in the genome of *Actinoplanes* sp. SE50/110. BlastP alignments against the MalP*^Ec/Cg^* amino acid sequences of *C. glutamicum* Cg1479 (43% identities, 60% positives) and *E. coli*-K12 JW5689 (46% identities, 39% positives) showed moderate sequence homologies to ACSP50_6911. In contrast, BlastP alignments against GlgP*^Cg^* sequences of *C. glutamicum* Cg2289 (25% identities, 39% positives) and *Streptomyces glaucescens* GLA.O SGLAU_23375 (25% identities, 38% positives) showed only weak sequence similarities. Therefore, we refer to it as putative maltodextrin phosphorylase MalP*^As^*. The putative MalP*^As^* protein (ACSP50_6911) was produced heterologously in *E. coli* BL21 (DE3) pLysS and its identity was confirmed by SDS-PAGE analysis and peptide fingerprinting using MALDI-TOF-MS/MS. For the detection of possibly cleaved glucose 1-phosphate, the coupled hexokinase/glucose 6-phosphate dehydrogenase enzyme assay was adapted. The hexokinase was exchanged for a phosphoglucomutase according to the protocol of Clermont et al. (2015) [18]. The phosphoglucomutase catalyzes the interconversion of glucose 1-phosphate and glucose 6-phosphate. Comparable to the coupled hexokinase/glucose 6-phosphate dehydrogenase enzyme assay, glucose 6-phosphate serves as a substrate for a glucose 6-phosphate dehydrogenase that leads to the reduction of NADP to NADPH. Analogous to previous assays, linear α-1,4-glucans from maltose to maltooctaose were used as substrates. Essentially, no activity was detected with maltose or maltotriose as substrates and only minimal activity was detected with maltotetraose as a substrate (Figure 3). However, substrates consisting of five or more α-1,4-linked glucose molecules resulted in an enhanced release of glucose 1-phosphate. The initial velocity was calculated from a glucose 1-phosphate calibration curve at the beginning of linearly increasing absorption. LC-ESI-MS data were collected to confirm the identity of the cleaved product. Glucose 1-phosphate (*m/z* 259.0213 [M − H]^−^) was confirmed as a product by the mass of the detected peaks and comparisons to a standard as well as a negative control with heat-inactivated enzyme (Appendix A).

Initial conversion is almost stagnant for substrates G5 to G7 (G5 = 41.69 ± 2.14 µM min^−1^, G6 = 42.57 ± 1.83 µM min^−1^, G7 = 42.31 ± 2.6 µM min^−1^) and slightly increases as the length increases (G8 = 46.64 ± 0.35 µM min^−1^) (Figure 3). Using glycogen as a substrate results in the fastest formation of glucose 1-phosphate (glycogen = 91.28 ± 1.67 µM min^−1^).

### 3.4. ACSP50_7587 Is a 4-Glucanotransferase and Functionally Similar to MalQ

According to the description of *E. coli*, MalP*^Ec^* and MalQ*^Ec^* are the most significant contributors to the maltose/maltodextrin pathway [11]. Upon analyzing protein sequence homology in databases of various organisms, significant similarities to 4-α-glucanotransferase (EC 2.4.1.25) were discovered in various *Actinoplanes* sp. and members of the Micromonosporaceae family. The percentage of identities ranged from 60% to 88%. The putative 4-α-glucanotransferase MalQ*^As^* (ACSP50_7587) recognizes linear maltodextrins and transfers an α-1,4-glucan segment to a reducing end of an acceptor molecule, commonly another α-1,4-glucan while releasing the remaining donor molecule [9]. Simultaneous assembly and degradation of unbranched glucans occurs via the maltose/maltodextrin metabolism. The cleaved glucose is phosphorylated to glucose-6P by glucokinase-mediated phosphorylation and delivered to glycolysis [9].

The activity and function of MalQ*^As^* in *Actinoplanes* sp. SE50/110 is not yet known in detail, especially since a homolog of the putative *malQ^As^* gene is found in the acarbose cluster (AcbQ; BlastP: 42% identity, 56% similarity) [4,6]. The putative *malQ^As^* gene was cloned into the inducible vector pJOE5751.1 and heterologously expressed *in E. coli* BL21(DE3) pLysS and the product purified via His_6_-tag. Protein identity was confirmed by SDS-PAGE and peptide fingerprint analysis. To determine the minimal substrate size, thin-layer chromatography (TLC) and glucose release assays were performed (Appendix A).

To analyze the distribution of the MalQ*^As^* products that were formed in in vitro assays, the extended glycan chains were measured using HPAEC-PAD. The concentration of glucans with chain lengths ranging from one to eight glucose molecules, designated as G1 to G8, was measured and calculated based on standard curves. The obtained results are presented in Figure 4, which includes all substrates and products for this specific standard range. As expected from pre-tests, each substrate resulted in the formation of elongated α-1,4-glucans. In the G2 substrate assay, approximately half of the substrate remains unhydrolyzed. Notably, there are high levels of G1 released, and relatively low levels of G3 to G5 produced. In pre-tests, G3 appeared to be the most favored substrate as the reaction rate was the highest. The product spectrum displayed, in addition to G1, G2, and probably unhydrolyzed G3, higher concentrations of G5, probably formed by the transfer of a cleaved maltosyl residue to a G3 acceptor molecule. In assays with larger substrates, elongated α-glucans, such as G6 and G7 were also detected. The product distribution shifted towards larger α-glucans in the third assay when almost 90% of G4 was used as a substrate, as the concentrations of G1, G2, and G3 decreased compared to the second assay. G5 had the highest concentration among the products. In the fourth assay 35% of the substrate G5 remained unhydrolyzed, whereas larger quantities of G6 to G8 were identified. Upon reviewing the chromatogram of this assay, new peaks with longer retention times emerged after the last standard G8 peak (Appendix A). These products are likely to have a range from G9 to G17.

#### MalQ^As^ Can Elongate Acarbose but Not Acarbose 7-Phosphate

It has been previously reported that *Actinoplanes* sp. SE50/110 produces longer acarviosyl metabolites as byproducts during acarbose production. This predominantly depends on the carbon source in the cultivation medium [8]. Given the involvement of AcbQ in elongating acarviosyl metabolites in the acarbose biosynthesis gene cluster and its sequence similarity to MalQ*^As^*, it was worthwhile to explore the potential role of MalQ*^As^* in producing these acarviosyl metabolites [4].

Analogous to these experiments, acarbose and acarbose 7-phosphate were tested alone and in combination with maltotriose as a second substrate. Acarbose and acarbose 7-phosphate were not or barely recognized as a substrate by MalQ*^As^* (Appendix A). The glucose release could be measured for the substrate combinations, which exhibited no relevant differences from the assays with maltotriose as the sole substrate (Appendix A). Another suggestion was that acarbose and acarbose 7-phosphate serve as an acceptor for glucanotransferase activities of MalQ*^As^* as well as for AcbQ [4]. To identify potentially newly formed acarviosyl metabolites, LC-ESI-MS analyses were performed. Besides acarbose, the masses detected were identified as acarviosyl-maltotriose (Ac-G3, *m/z* 808.317 [M + H]^+^), acarviosyl-maltotetraose (Ac-G4, *m/z* 970.36 [M + H]^+^) and acarviosyl-maltopentaose (Ac-G5, *m/z* 1132.41 [M + H]^+^) (Figure 5). Interestingly, no products of extended length were discovered in MalQ reactions with maltotriose and acarbose 7-phosphate (Appendix A).

## 4. Discussion

Since identifying maltose as the most effective carbon source for producing acarbose in *Actinoplanes* sp. SE50/110, understanding the mechanisms of maltose/maltodextrin metabolism and its potential connection to acarbose biosynthesis has become a focus of research. A first model presented by Schaffert et al. (2019) of the maltose/maltodextrin metabolism was established based on homology comparison to proteins of *E. coli* and *C. glutamicum* [6]. Here, this model along with in vitro assays and heterologously produced enzyme with predicted functions were utilized to expand the current model [6] (Figure 6).

In a previous study, ACSP50_4430 was predicted to be a maltodextrin glucosidase MalZ*^As^* [6]. Here, the α-glucosidase activity with maltotriose as a minimal substrate size was confirmed. Moreover, it has been shown acarbose has no inhibitory effect on MalZ*^As^* reactions and serves as a substrate for MalZ*^As^*. Similar effects have been reported for MalZ*^Ec^* in *E. coli* [3,14]. However, unlike in *E. coli*, acarbose is a potentially significant secondary metabolite for the survival strategy in *Actinoplanes* sp. SE50/110. In previous proteome and transcriptome dynamics data based on *Actinoplanes* sp. SE50/110 grown on maltose minimal medium, no MalZ*^As^* protein was detected throughout the cultivation, whereas AmlE was consistently abundant [28]. It is possible that regulatory effects enable the transcription and translation of MalZ*^As^* under different conditions. The acarviosyl unit is attached to the sugar, maltose in the case of acarbose, and its production depends on the available carbon source in the medium [8]. The impact of alternative carbon sources on MalZ*^As^* expression has not been explored, so it is possible that it could affect the hydrolysis of larger acarviosyl metabolites. These investigations are still pending. Considering the low activity of MalZ*^As^* compared to AmlE, a minor role in maltose/maltodextrin metabolism can be assumed for MalZ*^As^* but maybe in the hydrolysis of acarviosyl metabolites.

Initial studies have indicated that the α-glucosidase AmlE plays a critical role in the assimilation of maltose in *Actinoplanes* sp. SE50/110 [6]. The substrate spectrum was expanded beyond maltose to include linear α-1,4-glucans in general. The ability of AmlE to hydrolyze various glycosidic bonds led to the hypothesis that it may be involved in numerous carbohydrate assimilation processes. However, the deletion resulted in the absence of maltase activity within the crude protein extract, and a lack of growth on maltose occurred, indicating that the primary function involves maltose metabolism [6].

Here, we have verified the maltodextrin phosphorylase activity of MalP*^As^*, even though it was previously hypothesized that *Actinoplanes* sp. SE50/110 does not possess this activity. The substrate spectrum revealed that MalP*^As^* functions as both a maltodextrin and glycogen phosphorylase. Linear α-glucans with at least four glycosyl residues as well as glycogen serve as a substrate. In 2009, in *C. glutamicum*-only low-specific MalP*^Cg^* activities for substrates smaller than maltopentaose were described as well [18]. Furthermore, it was reported that the activities did not differ significantly when maltohexaose or maltopentaose was used as a substrate [18]. Due to the absence of a second homolog, MalP*^As^* in *Actinoplanes* sp. SE50/110 could potentially have a dual function in maltose/maltodextrin metabolism and glycogen degradation. The classification of ACSP50_6911 as MalP*^As^* or GlgP*^As^* is not entirely clear. In *E. coli*, MalQ*^Ec^* and MalP*^Ec^* are encoded in a single operon under the control of the maltose-dependent activator MalT*^Ec^*, whereas *glgP^Ec^* is localized in the *glgCAP* operon [29]. In *Actinoplanes* sp. SE50/110, both substrate spectra are covered by a single enzyme. After the initial classification of MalP*^Cg^* and GlgP*^Cg^* in *C. glutamicum*, the classification system of bacterial α-glucan phosphorylases is later questioned based on new findings that MalP*^Cg^* and GlgP*^Cg^* substrate spectra overlap [18]. The MalP*^Cg^* activity is regulated by the presence of ADP-glucose, a postulated feature of GlgP*^Cg^* [18]. Although glycogen and maltodextrin phosphorylases follow the same catalytic mechanisms and the catalytic domain seems to be highly conserved and they differ in regulation, there is no evident explanation for substrate preference, neither for glycogen nor linear maltodextrins [30]. It is postulated that binding branched oligosaccharides could cause a conformational change in the catalytic domain that may enhance affinity for glycogen [31]. Since no information is currently available regarding the regulation of MalP*^As^* in *Actinoplanes* sp. SE50/110 and it is not situated within any known operon, the naming MalP*^As^* is based on sequence homology comparisons with other organisms, wherein the similarities to MalP proteins are higher than to GlgP.

The putative MalQ^As^ protein was confirmed as a functional 4-glucanotransferase in *Actinoplanes* sp. SE50/110 and the substrate and product spectrum have been expanded beyond previous assumptions. In various MalQ*^Ec^* reactions, it is described that glucose is cleaved from the substrate and released [14]. The remaining glucose is transferred to the non-reducing end of other α-glucans. When considering the distribution of chain lengths of products based on the previous assumption that glucose is cleaved from an α-glucan with a chain length *n* and the remaining donor with a length *n* − 1 is transferred, the acceptor would always need to be extended by the unit *n* − 1. When maltotriose was used as the substrate, the acceptor was expected to be consistently extended by one maltose unit. As a result, substances such as maltopentaose and maltoheptaose might accumulate. However, it was also discovered that maltotetraose and maltohexaose were detected among the products, indicating that the cleavage is not specific to release only glucose. Studies of MalQ*^Ec^* by Weiss et al. in 2015 [13] demonstrated its ability to catalyze transglycosylation reactions in which glycosyl or dextrinyl units are transferred between linear maltodextrins of different lengths. These authors discovered that the equilibrium concentration of maltodextrin products is affected by the length of the initial substrate, with fewer glucose molecules released as the number of glycosidic bonds increased [13]. Similar conclusions could be drawn for product distribution of MalQ*^As^* in *Actinoplanes* sp. SE50/110.

The connection between maltose/maltodextrin metabolism and acarbose biosynthesis is of particular industrial interest. The dependence of the biosynthesis of acarviosyl metabolites on the carbon source has been studied, and it has been found that maltose-containing media leads to an increased yield of acarviosyl-maltose (acarbose) [8]. The sequence homolog of MalQ*^As^*, AcbQ was identified to be involved in the modification of acarbose [4]. The substrate spectrum of both enzymes overlaps within the group of linear α-glucans but differs in acarbose modification. AcbQ has the ability to synthesize multiple acarviosyl metabolites utilizing both acarbose and acarbose 7-phosphate as acceptor molecules [4]. Conversely, MalQ^As^ can only elongate acarbose. This mechanism may serve to prevent MalQ^As^ from being involved in acarviosyl metabolite formation.

It is noteworthy that AmlE is sensitive to acarbose but not to acarbose 7-phosphate, as revealed by the inhibitory effect of acarbose on its internal α-glucosidase. The potential physiological role of phosphorylating acarbose to acarbose 7-phosphate to prevent its degradation during the synthesis of elongated acarviosyl metabolites was already considered and supported in this work by the in vitro results of AmlE in this study [32].

Various acarviosyl impurities were identified during the industrial production of acarbose [20,33]. Initial successes have been achieved in eliminating impurities, such as component C [34]. However, the elongated acarviosyl metabolites formed by attached glycosyl units, which are produced by the MalQ homolog AcbQ, are also considered undesirable impurities. Reducing the accumulation of large α-glucans by knocking out *malQ^As^* or *acbQ* may prevent the production of elongated acarviosyl metabolites. The potential substrate spectrum of MalZ*^As^* within acarviosyl metabolites has not been fully characterized yet. However, since acarbose is a substrate, overexpression of *malZ^As^* may reduce elongated acarviosyl metabolites. Decreasing the biosynthesis of large α-glucans or glycogen could increase the substrate pool for acarbose biosynthesis.

## 5. Conclusions

This study deepens the understanding of the maltose/maltodextrin metabolism in the production of acarbose by *Actinoplanes* sp. SE50/110. Through the reconstruction of the pathway involving the enzymes AmlE, MalZ*^As^*, MalP*^As^*, and MalQ*^As^*, we elucidated their specific roles and reactions with different substrates. In particular, AmlE has a high hydrolysis rate for linear α-1,4-glucans and can hydrolyze different glycosidic linkages, whereas MalZ*^As^*, despite its slow hydrolysis activity on linear α-glucans, shows resistance to acarbose and can release glucose from acarbose. The dual function of MalP*^As^* as a maltodextrin and glycogen phosphorylase were highlighted, along with the central role of MalQ*^As^* in both degradation and synthesis of long-chain α-glucans. The findings demonstrate that MalQ*^As^* is not involved in the elongation of acarbose 7-phosphate, thus excluding it from the formation of acarviosyl impurities. This research provides a foundation for future studies aimed at a deeper understanding of the metabolic pathways of *Actinoplanes* sp. SE50/110 and enhancing acarbose production.

## Figures and Tables

**Figure 1 microorganisms-12-01221-f001:**
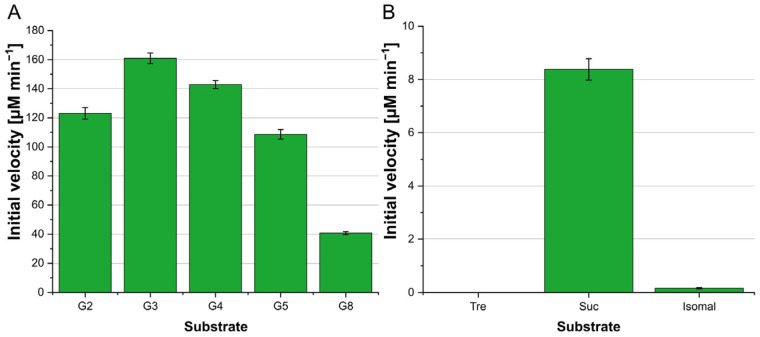
Characterization of the substrate spectrum of 1 µM of AmlE. NADPH formation from a hydrolytic enzyme assay was measured spectrophotometrically by absorption at 340 nm. Initial velocities were calculated from a standard calibration curve. (**A**) Major enzyme activity of AmlE. Substrates were maltose (G2), maltotriose (G3), maltotetraose (G4), maltopentaose (G5), and maltooctaose (G8). (**B**) Enzymatic site activity of AmlE. Used substrates were trehalose (Tre), sucrose (Suc), and isomaltose (Isomal). The negative control reaction was performed with heat-inactivated AmlE. All measurements were performed in triplicate (*n* = 3). Standard deviations are given.

**Figure 2 microorganisms-12-01221-f002:**
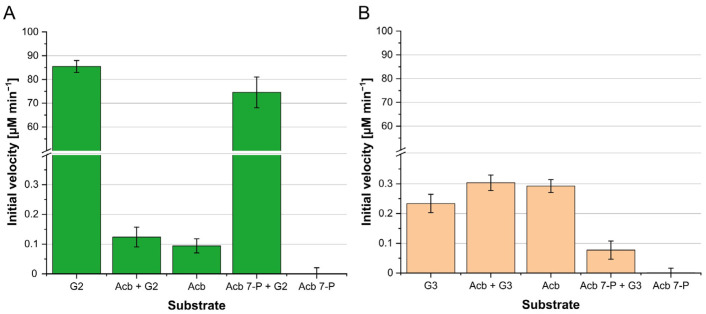
Characterization of the influence of acarbose on the hydrolysis of α-glucans of purified AmlE and MalZ*^As^*. The smallest determined substrate, respectively, maltose (G2) or maltotriose (G3), was used in combination with acarbose (Acb) or acarbose 7-phosphate (Acb 7-P). The NADPH release from a hydrolytic D-glucose enzyme assay was measured spectrophotometrically by absorption at 340 nm. Initial velocities were calculated from a standard curve. (**A**) 2 µM of purified AmlE, (**B**) 2 µM of purified MalZ*^As^*. Negative control reactions were performed with heat-inactivated enzyme. All measurements were performed in triplicate (*n* = 3) and standard deviations are indicated.

**Figure 3 microorganisms-12-01221-f003:**
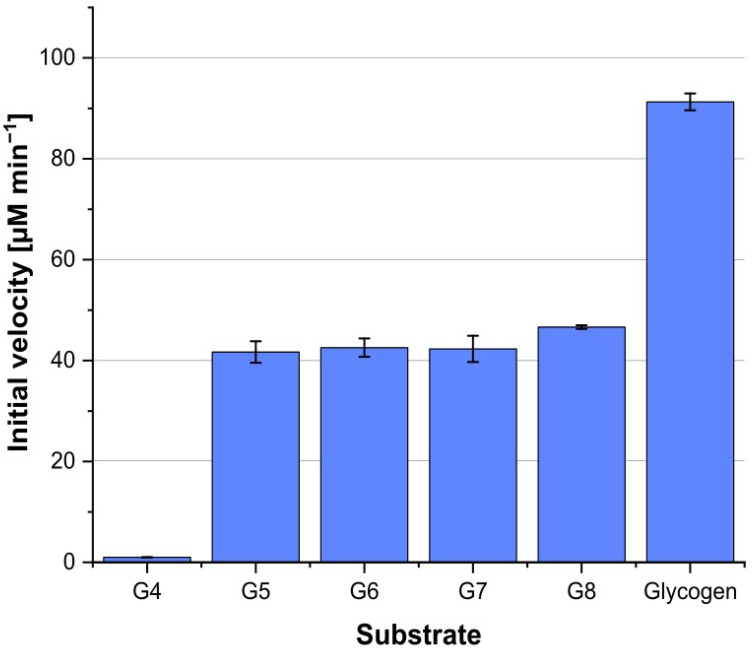
Characterization of the substrate spectrum of 1 µM of purified MalP*^As^*. NADPH release from a hydrolytic enzyme assay was measured spectrophotometrically by absorption at 340 nm. Initial velocities were calculated from a standard curve. Used substrates were maltotetraose (G4), maltopentaose (G5), maltohexaose (G6), maltoheptaose (G7) maltooctaose (G8), and glycogen. Negative control reaction was performed with heat-inactivated MalP*^As^*. All measurements were performed in triplicate (*n* = 3) and standard deviations are indicated.

**Figure 4 microorganisms-12-01221-f004:**
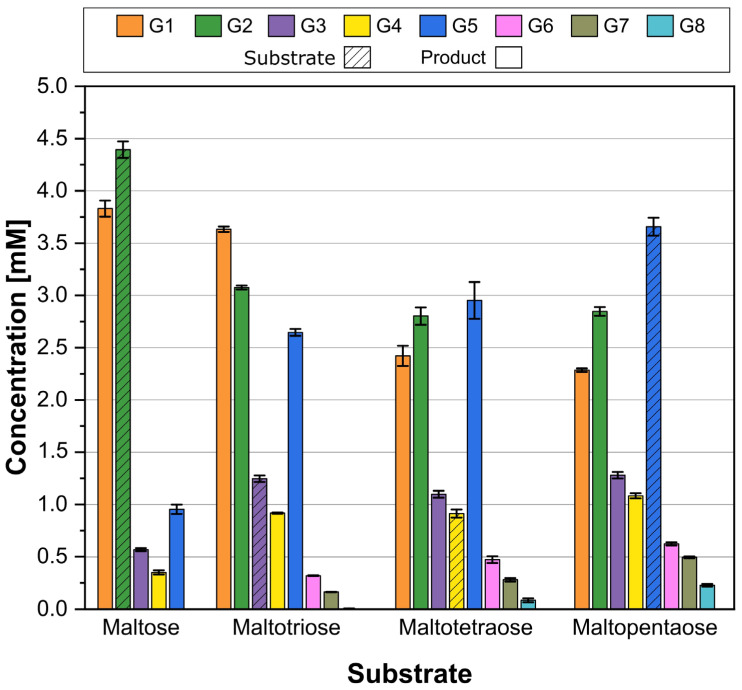
HPAEC-PAD analysis of MalQ^As^ reaction products resulting from in vitro enzyme assays. The substrates used were maltose, maltotriose, maltotetraose, and maltopentaose (dashed bars). Measured products were glucose (G1), maltose (G2), maltotriose (G3), maltotetraose (G4), maltopentaose (G5), maltohexaose (G6), maltoheptaose (G7), and maltooctaose (G8) (solid bars). Shown are the means of triplicate experiments (*n* = 3). Standard deviations are indicated.

**Figure 5 microorganisms-12-01221-f005:**
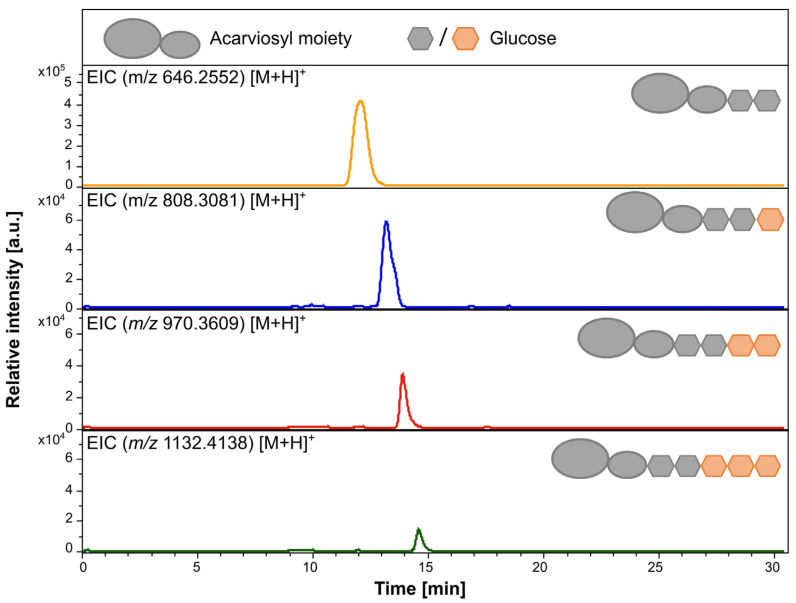
LC-ESI-MS analysis of MalQ*^As^* reaction products. In vitro reaction mixtures were incubated with acarbose and maltotriose as substrates. Elongated acarviosyl metabolites derived from acarbose were detected. The transferred glucose is illustrated as orange units. ESI (+) EIC for acarbose *m/z* 646.2553, ESI (+) EIC for acarviosyl-maltotriose *m/z* 808.3081, ESI (+) EIC for acarviosyl-maltotetraose *m/z* 970.3609, and ESI (+) EIC for acarviosyl-maltopentaose *m/z* 1132.4138.

**Figure 6 microorganisms-12-01221-f006:**
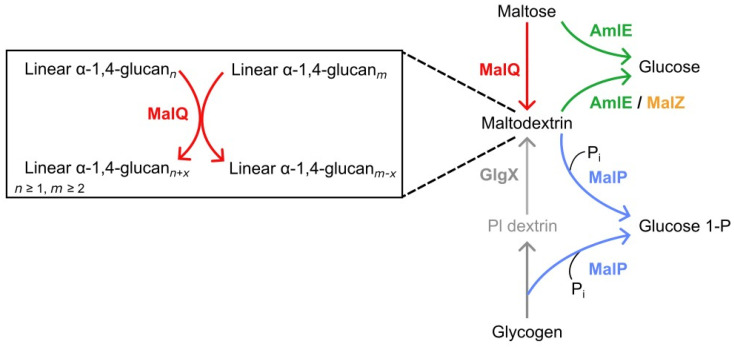
Updated pathway for the metabolism of maltose and maltodextrin in *Actinoplanes* sp. SE50/110 from the functional characterization of proteins produced heterologously in *E. coli*, and utilization of the model presented by Schaffert et al. [6]. The 4-glucanotransferase MalQ uses linear α-glucans with a minimal size of maltose as a donor molecule. It cleaves the reducing maltodextrinyl residue and transfers it to the non-reducing end of another α-glucan. The elongated α-glucan can then be used as an acceptor molecule to produce elongated maltodextrins. MalP degrades maltodextrins, provided by MalQ with a minimum length of five glycosyl residues, and supplies glucose 1-phosphate for metabolism. MalP serves a dual function as a phosphorylase for both maltodextrin and glycogen. Phosphatase-limited dextrins (Pl dextrin) are linearized by putative isoamylase-type debranching enzymes GlgX (ACSP50_2754, ACSP50_4618, ACSP50_6611). AmlE operates as a high-rate α-glucosidase and is mainly responsible for the intracellular assimilation of maltose and compensates for the low activity of MalZ.

**Table 1 microorganisms-12-01221-t001:** Bacterial strains used in this study.

Strain	Genotype	References
*E. coli* DH5α	F^–^ φ80*lac*ZΔM15 Δ(*lac*ZYA-argF)U169 *rec*A1 *end*A1 *hsd*R17(r_K_^–^, m_K_^+^) *pho*A *sup*E44 λ^–^*thi*-1 *gyr*A96 *rel*A1	Grant et al., 1990 [23]
*E. coli* BL21(DE3) pLysS	F^–^*omp*T *hsd*S_B_ (r_B_^–^, m_B_^–^) *gal dcm* (DE3) pLysS(Cm^R^)	Studier et al., 1986 [24]
*Actinoplanes* sp. SE50/110 (ATCC 31044)	wildtype	Frommer, 1979 [21]

## Data Availability

The original contributions presented in the study are included in the article/Appendix A, further inquiries can be directed to the corresponding author.

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
