# Peer review of "Role of MalQ Enzyme in a Reconstructed Maltose/Maltodextrin Pathway in *Actinoplanes* sp. SE50/110"

_microorganisms, 2024, doi:10.3390/microorganisms12061221_

Round 1
Reviewer 1 Report
Comments and Suggestions for Authors
The manuscript «Role of MalQ enzyme in a reconstructed maltose/maltodextrin pathway in Actinoplanes sp. SE50/110 » characterizes AmlE, MalZ, MalP, and MalQ enzymes that may be involved in maltose/maltodextrin metabolism through sequence homology comparisons, heterologous expression in E. coli and enzyme assays. The metabolism of α-1,4-linked glucose polymers in the maltose/maltodextrin system has been extensively researched in multiple microorganisms. The mal genes in Escherichia coli are regulated by MalT, a transcriptional activator that is induced by maltotriose and ATP and is crucial for the expression of the mal genes. However, in Actinoplanes sp. SE50/110 the metabolism of α-1,4-linked glucose polymers in the maltose/maltodextrin system and its participants is not sufficiently studied.
Actinoplanes sp. SE50/110 is the natural producer of acarbose, an α-glucosidase inhibitor that is used in diabetes type 2 treatment. The disaccharide maltose plays a unique role in the culture media as it serves as the primary carbon source, providing energy and serving as a key precursor in acarbose biosynthesis. MalQ is a 4-α-glucanotransferase, that cleaves linear maltodextrins. The maltose-induced α-glucosidase AmlE was deemed crucial for maltose metabolism in Actinoplanes sp. SE50/110. Authors show that the α-amylase AmlE from Actinoplanes sp. has the ability to hydrolyze a variety of glycosidic bonds and is sensitive to inhibition by acarbose. The putative maltodextrin glucosidase MalZ from Actinoplanes sp. catalyzes the hydrolysis of linear glucans containing three or more glycosyl units and uses acarbose as a substrate for slow release of glucose, but acarbose 7-phosphate did not lead to a release of glucose. It should be noticed that MalZ from Actinoplanes sp. has a low activity compared to AmlE. Here at first time was shown that Actinoplanes sp. MalP has a dual function as maltodextrin and glycogen phosphorylase in glycogen metabolism. Authors analyzed the distribution of the putative 4-α-glucanotransferase MalQ from Actinoplanes sp. (ACSP50_7587) products that were formed in in vitro assays and it was shown that MalQ from Actinoplanes sp. Is functionally similar to MalQ from E. coli. MalQ from Actinoplanes sp. can elongate acarbose but not acarbose 7-phosphate.
The manuscript is clear, relevant for the field and presented in a well-structured manner. The cited references include recent publications (within the last 5 years) and are relevant. It does not include an excessive number of self-citations. The manuscript scientifically sound and the experimental design is appropriate to test the hypothesis. The manuscript’s results could be reproduce based on the details given in the methods section. The figures and tables are appropriate and properly show the data.
Only one notice: there is absent the Conclusions section. There is widely Discussion section, but nevertheless to facilitate understanding of the results there should be at least a couple of sentences in Conclusions.
Author Response
Reviewer's comment: "Only one notice: there is absent the Conclusions section. There is widely Discussion section, but nevertheless to facilitate understanding of the results there should be at least a couple of sentences in Conclusions."
Our answer: Thank you for your helpful review. A conclusion section has now been added to the manuscript (lines 589 - 602).
Reviewer 2 Report
Comments and Suggestions for Authors
The manuscript is suitable for publication in Microorganisms.

Author Response
Thank you for taking the time to provide a comprehensive review of this manuscript. We have revised the text in accordance with your feedback, as outlined below:1) Line 24: Full stop has been added
2) Line 171: Second “analysis” has been eliminated
3) Figure 1 and 3: Concentrations of purified protein has been added
4) Figure S3: TLC analysis of MalQ activity as been added. Please be advised that as a consequence, the numbering of the following figures has been adjusted.
5) Abbreviations of the Journal Names were homogenized manually. Please be advised that we have used the automatic MDPI citation style and therefore hope that the Citavi plug-in will not automatically overwrite these changes.